# Influence of Lactitol and Psyllium on Bowel Function in Constipated Indian Volunteers: A Randomized, Controlled Trial

**DOI:** 10.3390/nu11051130

**Published:** 2019-05-21

**Authors:** Jing Cheng, Julia Tennilä, Lotta Stenman, Alvin Ibarra, Mandhir Kumar, Kamlesh Kumar Gupta, Shyam Sundar Sharma, Dhiman Sen, Sandeep Garg, Mukund Penurkar, Santosh Kumar, Arthur C. Ouwehand

**Affiliations:** 1DuPont, Global Health & Nutrition Science, Danisco Sweeteners Oy, Sokeritehtaantie 20, FI-02460 Kantvik, Finland; jing.cheng@dupont.com (J.C.); julia.tennila@outlook.com (J.T.); lotta.stenman3@gmail.com (L.S.); arthur.ouwehand@dupont.com (A.C.O.); 2Department of Nephrology, Sir Ganga Ram Hospital, Old Rajindra Nagar, New Delhi 110060, India; mandhirkr@yahoo.co.in; 3Department of Medicine, King George’s Medical University Chowk, Lucknow 226003, Uttar Pradesh, India; kamleshkgmu@rediffmail.com; 4Department of Gastroenterology, Sawai Man Singh Hospital. JLN Marg, Jaipur 302004, Rajasthan, India; shyamsharma4@rediffmail.com; 5Department of Clinical Trial & Research, Apollo Gleneagles Hospitals, Kolkata 58, Canal Circular Road, Kolkata 700054, West Bengal, India; dhimansen2004@yahoo.com; 6Department of Medicine, Maulana Azad Medical College & Associated Lok Nayak Hospital, Bahadur shah Zafar Marg, New Delhi 110002, India; drsandeepgargmamc@gmail.com; 7Department of Medicine, Sanjeevan Hospital, Plot No 23, Off. Karve Road, Erandwane, Pune 411 004, India; drpenurkarm@gmail.com; 8Department of SAS & Biostatistics – JSS Medical Research, JSS Medical Research India Private Limited, Mathura Road Sector 27 D, Faridabad, Haryana 121003, India; santosh1.kumar@jssresearch.com

**Keywords:** constipation, lactitol, psyllium, randomized clinical trial

## Abstract

Psyllium and lactitol have been reported to increase fecal volume, moisture content and bowel movement frequency (BMF). However, the benefits of their combined use on constipation has not been examined. The aim of this study was to evaluate the effects of a 4-week intervention with lactitol and/or psyllium on bowel function in constipated volunteers. Adults (*N* = 172) who were diagnosed with functional constipation per Rome III criteria were randomized to four treatment groups: 10 g lactitol, 3.5 g psyllium, a combination of 10 g lactitol and 3.5 g psyllium, or placebo. The primary endpoint was change in BMF from Day 0 to 28 as compared to placebo. Secondary endpoints were assessed by inventories, including stool consistency, patient assessment of constipation symptoms and quality of life, relief of constipation, 24-h food recall, physical activity, product satisfaction and adverse events (AE). BMF increased by 3.0 BMs with lactitol, by 2.9 with psyllium, and by 3.1 with the combination, but was not different from placebo (3.7 BMs). Other clinical endpoints were similar between treatments. No serious AEs were reported. In conclusion, this study showed a similar effect on relief of constipation in all treatment groups. The treatments that were administered to the volunteers were well tolerated.

## 1. Introduction

Constipation, functional constipation, and chronic constipation are often used interchangeably, although the latter includes conditions that are caused by diseases, such as neurogenic and drug-induced diseases [1]. In general, constipation is one of the most common gastrointestinal complaints, affecting 14% to 16% of the general population worldwide [2]. It is associated with a significantly impaired quality of life, psychological distress [3], increased health care costs, and lower work productivity [4]. Its prevalence increases with age and is more frequent in females and in Western populations [2,5].

Per the Rome III diagnostic criteria for functional constipation in adults, the complaint is defined as unsatisfactory defecation that is characterized by infrequent stools (fewer than 3 defecations per week) and difficult stool passage, with the presence of at least 2 of the following symptoms for at least 25% of defecations: straining, a sense of difficulty passing stool, incomplete evacuation, hard/lumpy stools, prolonged time to stool, and the need for manual maneuvers to pass stool [6]. Further, these criteria should have been met for the past 3 months, with symptom onset at least 6 months prior to diagnosis [6]. In addition, loose stools should be rare, without laxative use, and the symptoms should not fulfill the diagnostic criteria for irritable bowel syndrome. 

The multisymptomatic, heterogeneous nature of constipation is a major hurdle in the management of subjects with this condition [6]. Treatment of constipation continues to evolve and remains challenging. There is a considerable range of treatment modalities that are available for subjects with constipation, but the clinical evidence that supports their use varies widely [7]. The initial therapy includes recommendations for lifestyle modifications, such as adequate fluid intake, non-strenuous exercise, and increased dietary fiber intake [8,9]. Traditional methods for treating constipation include bulking agents (e.g., psyllium, methylcellulose, bran, and calcium polycarbophil), stool softeners (e.g., docusate sodium), stimulant laxatives (e.g., senna, bisacodyl), and osmotic laxatives (e.g., milk of magnesia, sorbitol, lactulose, lactitol, and polyethylene glycol) [10]. Constipation due to organic causes often results in persistent symptoms under traditional treatments; thus, novel therapies, such as prokinetics (e.g., prucalopride) and secretory agents (e.g., lubiprostone and linaclotide) are applied as targeted therapies [1,9]. Recently, probiotics have garnered increased interest as an adjuvant therapy for alleviating the symptoms of constipation and maintaining health [11,12,13]. 

Bulking agents are generally considered the first-line therapy for constipation. These agents expand with water to increase stool bulk and enhance bowel movements. Adequate fluid intake is necessary for bulk-forming agents—a lack of water increases bloating and, paradoxically, can increase the incidence of bowel obstruction [14]. Among bulking agents, psyllium relieves constipation, and several randomized clinical trials have demonstrated its benefits in improving bowel function [15,16,17,18].

In subjects who are unresponsive or intolerant to bulking agents, osmotic laxatives may be recommended [19]. Osmotic laxatives are widely recommended in the management of constipation, based on their excellent efficacy, tolerability, and safety in pediatric subjects, during pregnancy, and for mature adults [20,21]. These agents adsorb and retain water due to their hyperosmolar nature and enhance stool passage [14].

Lactitol, a synthetic disaccharide osmotic laxative, is produced by the hydrogenation of lactose. It increases osmotic pressure in the intestinal lumen, resulting in greater fecal volume and moisture content and the stimulation of peristalsis [22]. The efficacy of lactitol in the treatment of constipation has been established by various clinical trials in many subjects, ranging from children to the elderly [23,24,25,26]. Several systematic reviews have summarized the efficacy, tolerability, and palatability of lactitol [27,28]. Few clinical trials have reported greater efficacy with a combination of laxatives (senna and psyllium) for constipation [29,30,31]. However, the benefits of the combination lactitol and psyllium in cases of functional constipation have not been examined. Therefore, this study determined the efficacy and safety of lactitol and psyllium individually and in combination in subjects with constipation.

## 2. Materials and Methods 

This prospective, randomized, 4-arm, double-blind, placebo-controlled, phase III intervention study was performed in 6 clinics throughout India (Appendix A). DuPont, Global Health & Nutrition Science, Danisco Sweeteners Oy (Kantvik, Finland), coordinated the study and provided the investigational products (Danisco USA Inc., Madison, WI, USA). JSS Medical Research India Private Limited (Haryana, India)—formerly known as Max Neeman International—oversaw the trial coordination, monitoring, data collection, and statistical analysis.

### 2.1. Study Volunteers

The study volunteers were healthy adults (aged 25–65 years) with self-reported constipation (≤3 stools per week for the last 3 months and in the run-in period), attending local practitioners for bowel function related disorders. Other inclusion criteria were body mass index (BMI) between 18.5 (inclusive) and 29.9 (inclusive), at least 80% compliance during the run-in period, willingness to maintain a stable diet throughout the study, and the ability to comprehend the study, including the possible risks and side effects that are associated with both investigational products (IPs) and constipation-associated medications (e.g., rescue medications). Blood and urine tests for safety were performed at the screening visit, and volunteers without clinically significant findings were considered eligible.

Volunteers were excluded if they had consumed probiotics, prebiotics, laxatives, fiber supplements, or other medications for constipation that could have affected the study outcome within 2 weeks prior to the screen and during the run-in period. Other exclusion criteria were diagnosed or suspected organic gastrointestinal disease (e.g., colitis, Crohn’s disease, irritable bowel syndrome, celiac disease, bowel surgery) or severely impaired general health, history of alcohol or drug abuse, and participation in another clinical trial within 3 months prior to the screen. Volunteers who were planning major lifestyle changes (e.g., diet, exercise level, travel), were pregnant, were planning a pregnancy or breastfeeding, or had given birth in the last 3 months were also excluded.

Medications that excluded participation from the study included antibiotics and oral steroids (1 month prior to the start of the study) and current use of anticholinergic medications, pain medications that contained opiates or morphine, weight loss medications, misoprostol, 5-HT3 receptor antagonists, antacids with magnesium or aluminum, diarrhea medication, medication that accelerated emptying of the stomach, sulfasalazine, cholestyramine, cytostats, and biological medications or probiotics, because they could have affected the study outcomes. Iron, antidepressants, statins, thyroxine, coxibs, acid medications, inhaled steroids, and other non-excluding medications that would not be expected to affect the study outcomes in the clinician’s opinion were allowed if they had been consumed at least 30 days prior to the study on a stable dose; otherwise, volunteers who failed to meet the criteria were excluded. Volunteers who were unsuitable for the study for any other reason, as judged by the clinician, or had known allergies to any substance in the study product, were excluded.

### 2.2. Study Design

Eligible volunteers were enrolled in a 6-week intervention study, comprising a 2-week run-in period and a 4-week treatment period (Figure 1). A window of ±2 days was allowed in the intervention in case the volunteers were not able to attend the study visit at exact planned date. During this window, the protocol of the on-going intervention remained in force. Volunteers were instructed to take 1 sachet of an IP, mixed in 250 mL water, preferably during breakfast, daily for the entire intervention. During the first 2 weeks (run-in-phase), only placebo was administered (16 sachets/volunteer), and adherence was measured at baseline. Volunteers with at least 80% adherence at baseline and 3 bowel movements or less per week (see below) were enrolled in the study, randomized to 1 of 4 treatment groups, and given sachets of the IP accordingly.

At the end-of-study (EOS) visit on Day 28, volunteers returned empty and unused sachets of IPs for accountability. Rescue medications, including suppositories (bisacodyl or glycerin) and enemas, were permitted at the discretion of the investigator. Concomitant treatment for all groups—including iron, antidepressants, statins, thyroxine, coxibs, acid medications, inhaled steroids, and other non-excluding medications that did not affect the study outcomes in the clinicians’ opinions—was allowed if it had been consumed for at least 30 days on a stable dose. 

### 2.3. Methods

Diagnoses were made by the investigators, based on ROME III criteria for functional constipation in adults, during Visits 1 and 2 at the clinics.

The primary outcome was the change in weekly BMF from baseline to the end of the 4-week treatment. BMF was recorded daily by the volunteers in a diary throughout the study.

The secondary outcome of Stool Consistency (SC) was assessed using the Bristol Stool Form Scale [32], in which stool types are classified into 7 categories, from type 1 (separate hard lumps, constipation) to type 7 (liquid consistency, diarrhea). Stool consistency was recorded daily by the volunteers in a diary throughout the study. Other secondary outcomes were measured using validated questionnaires. Constipation symptoms and quality of life were evaluated with the validated Patient Assessment of Constipation Symptoms (PAC-SYM) [33] and Quality of Life (PAC-QoL) [34] questionnaires, for which the data were collected at baseline (Visit 3) and after 2 weeks (Visit 4) and 4 weeks of treatment (Visit 5, EOS) (Figure 1). The data were analyzed as total scores and as subscales on abdominal, rectal, and stool symptoms for PAC-SYM. For PAC-QoL, we analyzed the subscales on worries and concerns, physical discomfort, psychosocial discomfort, and satisfaction. Food intake was evaluated by 24-h food recall at baseline and after 4 weeks of treatment to quantify total calories (Kcal), fat (g), carbohydrates (g), protein (g), fiber (g), and liquid (mL). The food recall interview took place on a weekday (Tuesday through Friday). Each food item was quantified in terms of calories, macronutrients, and fiber using the Nutritive Value Chart of Indian Food [35]. Additionally, physical activity was evaluated with the International Physical Activity Questionnaire (IPAQ) [36] at baseline and after 4 weeks of treatment. The data were analyzed as a combined score and as subscales on walking and moderate and vigorous activity.

### 2.4. Study Treatment

The IPs were powder sachets that contained 10 g lactitol, 3.5 g psyllium, a combination of 10 g lactitol and 3.5 g psyllium, and placebo. Sucrose, instant waxy maize starch, citric acid, trisodium citrate, carotene powder, and orange flavor were used as excipients and as placebo. The IPs were produced by Danisco USA Inc. The products complied with the microbial quality standards per the European Pharmacopoeia. All 4 study treatments were as similar in smell, taste, and appearance as possible. The formulations of the IPs are listed in Appendix A.

### 2.5. Compliance Testing

At randomization (Figure 1), all randomized volunteers received IP (30 sachets/volunteer) according to the assignment. At the EOS visit, volunteers returned empty and unused sachets of the IP for accountability calculation.

### 2.6. Ethical Considerations

This study was conducted in full accordance with the Sixth Revision (2008) of the Declaration of Helsinki; the EMA Note for Guidance on Good Clinical Practice (GCP; CPMP/ICH/135/95 - in operation 17.01.97), also known as the International Council for Harmonisation of Technical Requirements for Pharmaceuticals for Human Use (ICH) guidelines for GCP; and the laws and regulations on clinical research in the participating countries. A notification was submitted to the national authority; i.e., the Drug Controller General of India (DCGI), before commencement of the trial on the 26 June 2014, as applicable according to local regulations. Written informed consent was obtained from each volunteer before any study-specific procedure or assessment was performed. The study was reviewed and approved by six local ethical committees of the research sites in India. The study was registered in the Clinical Trials Registry for India (WHO approved Registry) with the registration identification number CTRI/2015/02/005580. The EC approval numbers per research site are listed below:New Delhi, ECR/20/Inst/DL/2013/RR-16Lucknow, ECR/262/Inst/UP/2013/RR-16Jaipur, ECR/26/Inst/RJ/2013/RR-16Kolkata, ECR/373/Inst/WB/2013/RR-16New Delhi, ECR/329/Inst/DL/2013/RR-16Pune, ECR/54/Inst/Maha/2013/RR-16

### 2.7. Statistical Analyses

#### 2.7.1. Determination of Sample Size

The trial hypothesis was to obtain a significant difference between the 3 treatments in parallel against placebo. Thus, the significance level for each comparison was assumed to be 0.05/3 = 0.0167. The sample size was based on the comparison of change in bowel movements per week from baseline to EOS between the 3 treatment groups and placebo. Based on the literatures, a standard deviation of 3–5 for bowel movements per week and an effect size of 2–4 more bowel movements per week after treatment were assumed [25,37,38,39,40,41,42]. Taking into account a discontinuation rate of approximately 10%, 43 volunteers per treatment group—i.e., 172 volunteers in total—were recruited to obtain approximately 40 volunteers per treatment group, providing 80% power for statistically significant changes in bowel movements/week.

#### 2.7.2. Analysis Sets

The intention-to-treat (ITT) population was defined as all randomized subjects. The per-protocol (PP) population was defined as those subjects in the ITT population who had completed the EOS visit and for whom there were no serious protocol deviations (e.g., use of systemic or oral antibiotics, study product compliance below 80%, missing study diary recordings from critical days of measurement, constipation- or diarrhea-related medications). Moreover, the PP population should have less than three Bowel Movements (BM) per week during the run-in period, based on a self-reporting participant’s diary, as defined during the Blind Data Review process.

#### 2.7.3. Randomization

Blocked randomization was stratified in order to include both vegetarian and non-vegetarian subjects at equal ratio. Subjects were randomized 1:1:1:1 to one of the four treatment groups with the block size for 4. The randomization lists were prepared by Oy 4Pharma Ltd. (Turku, Finland) using a validated computer program, SAS^®^ for Windows version 9.3/Proc PLAN (SAS^®^ Institute Inc., Cary, NC, USA). The sealed envelopes containing randomization codes were sent directly to JSS Medical Research India Private Limited. Mechanism of concealment was to use 1280 random numbers between 0001-9999 for kits with prefix “K” (e.g., K3625). These numbers were randomly assigned to the subject numbers used over the study and merged with the output of the randomization for the subjects. This merged output of randomization was automatically written in an output text file which was read in to MS Word to construct a Randomization List. The subject number, associated treatment sequence and kit number were provided within each stratum in the Randomization List. The kit (number) linked to each subject and subject’s randomized treatment sequence contained the treatment randomized for the subject. All subjects and investigators were masked to the treatment assigned until completion of the trial to minimize any subject- or investigator-induced bias in the outcome measures. 

#### 2.7.4. Analysis of Efficacy

The efficacy was analyzed in the ITT population. Missing values were imputed by the last-observation-carried-forward approach.

The descriptive statistics for continuous variables was presented as number (*n*) of non-missing observations, mean, standard deviation, median, and minimum and maximum (range). For categorical data, the descriptive statistics were presented as the number of exposed patients and number (*n*) with percentage of observations in various categories of the variable, for which the percentage was based on the exposed patients. Unless otherwise stated, all statistical tests were two-sided hypothesis tests, performed at a 5% level of significance for the main effects, and all confidence intervals were two-sided 95% confidence intervals. Descriptive analyses also included graphical presentations of the data, wherever appropriate. 

Comparisons were made using chi-square test or Fisher’s exact test for categorical variables and paired-t test was used for comparisons within treatment groups; analysis of covariance (ANCOVA) was performed with treatment, population strata, and research center as factors and baseline as the covariate for treatment comparisons for continuous variables. Dunnett’s test was used for group-wise comparisons (treatments versus placebo). All tests were two-tailed, and the significance level was 0.05.

All statistical analyses were performed at JSS Medical Research India Private Limited using SAS^®^ Version 9.4 package (SAS^®^ Institute Inc., Cary, NC, USA).

#### 2.7.5. Analysis of Safety

All randomized volunteers were defined as the safety population. Adverse events (AEs) were coded by system organ class (SOC) and their preferred terms using the Medical Dictionary for Drug Regulatory Affairs (MedDRA) (v. 19.0 or later). All AEs were listed and categorized as AEs before dosing and treatment-emergent adverse events (TEAEs), i.e., AEs with onset on the date of or after dosing. TEAEs were tabulated by indicating the number and percentage of volunteers and the number of events. The number of volunteers who experienced any adverse drug reaction, including any adverse drug reaction, was summarized for each treatment group. AEs were collected, evaluated, and tabulated in relation to study drug, seriousness, severity, action taken, outcome, SOC, and preferred term for each treatment group. Serious adverse events (SAEs) were summarized by SOC and their preferred term. 

The AEs that were related to severity were presented as number of occurrences and % for the respective group. The AEs and their relationship with the study product were presented as above or by severity. The SAEs that were related to seriousness were expressed as the number of occurrences and % for the respective group. Clinical laboratory parameters (blood cytology, blood chemistry), vital signs (pulse rate, systolic and diastolic BP, and body temperature), and physical examination findings were measured descriptively. AEs were recorded in a daily diary throughout the intervention.

All statistical analyses were performed at JSS Medical Research India Private Limited using SAS^®^ Version 9.4 package (SAS^®^ Institute Inc., Cary, NC, USA).

## 3. Results

From April 20, 2015 to April 28, 2016, 204 volunteers were screened, out of which 172 were enrolled. Their disposition is shown in Figure 2. There were 5 reasons for non-inclusion/screen failure (*N* = 32): self-reported constipation >3 stools per week (*N* = 13, 40.6%), noncompliance with the study product and methods (*N* = 9, 28.1%), deviations from the protocol (*N* = 7, 21.9%), failure in the safety tests (*N* = 2, 6.3%), and less than 80% compliance with the treatment during the run-in period (*N* = 1, 3.1%). All volunteers attended all visits; the flowchart is presented in Figure 2.

### 3.1. Demographics and Baseline Characteristics

Table 1 shows the demographics and baseline characteristics of the volunteers in the ITT population. In general, the enrolled volunteers were middle-aged (mean age: 40.3 ± 10.5 y), normal-weight adults (mean BMI: 23.5 ± 2.5), with similar rates of males (*N* = 80, 46.5%) and females (*N* = 92, 53.5%). Although significant differences were observed in gender, height, and BMI compared with placebo, they were not considered to have had any influence on the clinical endpoints in the efficacy analysis. Overall, the demographics were comparable across the 4 treatment groups. 

### 3.2. Clinically Relevant Medial History and Concomitant Medication

Of 172 randomized volunteers, 39 reported a clinically relevant medical history but were evenly distributed across all treatment arms (Appendix A). Over 1% of the volunteers in the ITT population were on concomitant medications during the treatments (Appendix A), including vitamins (2.3%); drugs for acid-related disorders (1.7%); and drugs for functional gastrointestinal disorders, mineral supplements, and analgesics (1.2%). In the lactitol group, 1 volunteer took paracetamol during the study. In the psyllium group, 3 volunteers took vitamins, 2 took minerals, and 1 took vitamin B. In the combination product group, 1 volunteer each was on paracetamol, nimesulide, esomeprazole sodium, bisacodyl, and calcium, and 2 volunteers each were on pantoprazole sodium sesquihydrate and drotaverine hydrochloride. In the placebo group, 1 volunteer each was on omeprazole and calcium carbonate, cholecalciferol.

### 3.3. IP Compliance

No statistically significant difference was observed in total consumed sachets between active treatment groups versus placebo (Table 2). Overall, the 172 randomized volunteers were at least 85% compliant with the study treatment in all groups.

### 3.4. Efficacy

Table 3 summarizes the results of the clinical evaluations. The analyses were performed in the ITT population, which was randomly assigned to 4 treatment groups.

The primary endpoint was the mean change in BMF between pre-treatment and post-treatment compared with placebo. There was a significant increase in BMF from baseline to EOS in all 4 treatment groups, with all treatments effecting an improvement in constipation over the 4-week intervention (*p* < 0.01, Table 3). A significant increase in BMF was also noted from baseline to Days 7, 14, and 21 in each treatment group in the ITT population (Appendix A). However, no significant difference in BMF was observed between the 4 treatment groups or for any active treatment versus placebo. Similar results were observed for several secondary endpoints, such as SC, PAC-SYM, and PAC-QoL.

There was no significant increase in walking or moderate, vigorous, or total physical activity in the 3 active treatment groups from baseline to EOS. Notably, from baseline to EOS, a significant (*p* = 0.02) rise in total physical activity was observed in the placebo group (3345.5 ± 7228.1 MET-min/Week vs. 4124.1 ± 8254.9 MET-min/Week, respectively). This tendency was seen in the placebo group for walking activity (1358.1 ± 1776.7 MET-min/Week vs. 1861.4 ± 2253.0 MET-min/Week, respectively; *p* = 0.05).

Adequate relief of constipation (ARC) was measured at 2 site visits: baseline and EOS, with respect to a response of “ARC since last visit.” The proportion of volunteers with ARC was comparable between all treatment groups at baseline and EOS. Most volunteers (≥80%) without ARC during the run-in period reported ARC at EOS in all treatment groups.

A higher proportion of volunteers in all 4 treatment groups were ‘satisfied’ with the product (47.6% to 67.5%), followed by ‘very satisfied’ (7.5% to 23.8%), ‘neutral’ (12.5% to 17%), ‘dissatisfied’ (8.5% to 12.5%), and ‘very dissatisfied’ (0% to 4.7%). On all subscales, the proportion of those who experienced product satisfaction was comparable between all active treatments and between active groups versus placebo. 

No significant change in total calories (Kcal), carbohydrate (g), fat (g), protein (g), fiber (g), or liquid intake (mL) was observed from baseline to EOS in any treatment groups, except for a significant increase in fat intake and a decline in liquid intake from baseline to EOS in the placebo group (fat intake: 36.4 ± 16.3 g vs. 41.7 ± 18.6 g, *p* = 0.046; liquid intake: 1061.5 ± 827.0 mL vs. 921.3 ± 666.5 mL, *p* = 0.04) (Appendix A). In addition, a significant decrease in liquid intake was noted in the lactitol group from baseline to EOS (1104.3 ± 984.2 mL vs. 862.8 ± 498.0 mL, *p* = 0.02). Fiber intake did not change significantly in the treatment groups but trended toward a decline in the lactitol and combination groups (lactitol: 3.9 ± 1.8 vs. 3.3 ± 1.5 g, *p* = 0.09; combination: 4.2 ± 2.4 vs. 3.6 ± 1.6 g, *p* = 0.054). 

The changes in total calories, macronutrients, and liquid intake were comparable between active treatments versus placebo. A trend was observed towards a decrease in fiber intake (*p* = 0.06) from baseline to EOS in the lactitol versus placebo group.

### 3.5. Safety Results

A total of 14 AEs occurred in 9 (5.2%) volunteers (Appendix A): 6 AEs in 3 (6.4%) volunteers in the lactitol group, 1 AE in 1 (2.4%) volunteer in the psyllium group, 5 AEs in 4 (9.3%) volunteers in the combination group, and 2 AEs in 1 (2.5%) volunteer in the placebo group. Of these AEs, 13 were mild and 1 was moderate. Moreover, 13 AEs in 9 (5.2%) volunteers were unrelated to the study drug, and 1 AE in 1 (2.3%) volunteer in the combination group had a potential relationship with the treatment. 

The most common TEAEs were gastrointestinal disorders; 4 TEAEs in 3 (1.7%) volunteers, which were abdominal pain; 2 TEAEs in 2 (1.2%) volunteers, and pain in the extremities; 2 TEAEs in 1 (0.6%) volunteer. Seven TEAEs in 5 (2.9%) volunteers were unrelated to the study drug, and 1 TEAE in 1 (2.3%) volunteer in the lactitol 10 g + psyllium 3.5 g group was possibly related. No specific action was deemed to have been needed against any AE. 

## 4. Discussion

In this prospective, randomized, double-blind, placebo-controlled, parallel-group study, a statistically significant increase in BMF was noted in all 4 treatment groups from baseline to Days 7, 14, and 21 and from baseline to EOS (Day 28) (Table 3 and Appendix A). Fibers consistently increase BMF compared with placebo [43], which was not observed in our study, but we observed a significant increase from baseline. However, consistent with our findings, Ewerth et al. reported no difference of BMF between psyllium and placebo, but contrary to our findings, Ewerth et al. did not observe a significant change from baseline to EOS for both study products [44]. Further, several studies have shown a nonsignificant increase in BMF with such agents as bran (bulk), fiber, psyllium, and cisapride compared with the control group (regular diet or placebo) [45,46,47,48]. These studies were performed in various western countries around the world, while the present study was performed in India. Although it cannot be ruled out, it is not likely that geography explains the difference in functionality of the tested products.

To rule out the possibility of effects from the ingredients in the placebo, we assessed BMF during the run-in phase. As seen in Appendix A, the increase from the run-in period to the first week of treatment for placebo is unexpected, whereas the stability of BMF during the run-in phase indicates that placebo (which was consumed during the run-in) does not affect bowel function. Thus, we are unable to explain the cause of the undifferentiated efficacies between the treatments and placebo.

The Patient Assessment of Constipation Symptoms (PAC-SYM) and assessment of constipation Quality of Life (PAC-QoL) questionnaires are the best validated and most specific tools for measuring the quality of life of patients with constipation. These questionnaires were developed, based on the literature, clinical expert interviews, and patient interviews, to measure health-related quality of life in constipated patients [49]. In our study, total PAC-SYM and PAC-QoL scores declined significantly in the treatment groups from baseline to Day 28 in the ITT population, suggesting an improvement in the specific symptoms and QoL of patients in the study cohort. 

Additionally, no significant difference was observed in product satisfaction in any active treatment group compared with placebo in the ITT population, indicating no superior satisfaction with the active products versus placebo. A significant increase in fat intake and a decline in liquid intake were seen in the placebo group from baseline to Day 28 in the ITT population.

Our study products were well tolerated by the study patients. It is reported that both lactitol and psyllium are accompanied by uncommon adverse reactions. Lactitol, like most sugar alcohols, causes cramping, flatulence, and diarrhea, because humans lack a suitable beta-galactosidase in the upper gastrointestinal (GI) tract, and most ingested lactitol reaches the large intestine [22]. Psyllium husk is also well tolerated, although mild side effects can occur during treatment. The most common side effects include bloating and excess gas, which can be accompanied by abdominal cramping, nausea, and stomach pain [50]. 

A possible limitation of the study is that we failed to record the potential use of the traditional ayurvedic remedies for constipation relief. Another potential shortcoming is that rescue medication use was not defined, and therefore, this aspect was left to the judgement of the principal investigators. 

## 5. Conclusions

Overall, BMF increased similarly in all 4 treatment groups. The treatments that were administered to the subjects were well tolerated. All 4 treatments were safe, and no adverse reactions required further action. No serious AEs or TEAEs were reported. All laboratory findings and vital signs were comparable across all treatment groups.

## Figures and Tables

**Figure 1 nutrients-11-01130-f001:**
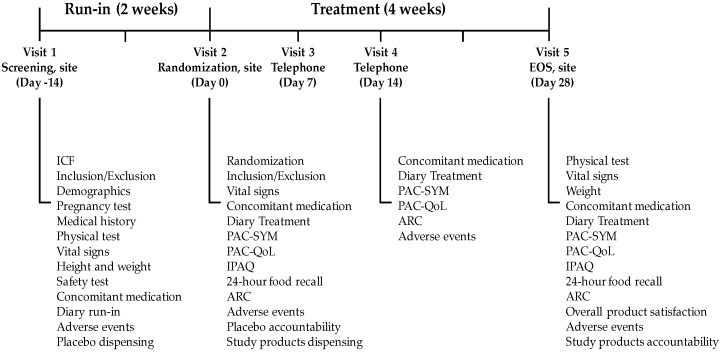
Study outline. There were 5 study visits during the entire study. EOS-End of Study; ICF-informed consent form; PAC-SYM-patient assessment of constipation symptoms; PAC-QoL-patient assessment of constipation quality of life; IPAQ-international physical activity questionnaire; ARC-adequate relief of constipation.

**Figure 2 nutrients-11-01130-f002:**
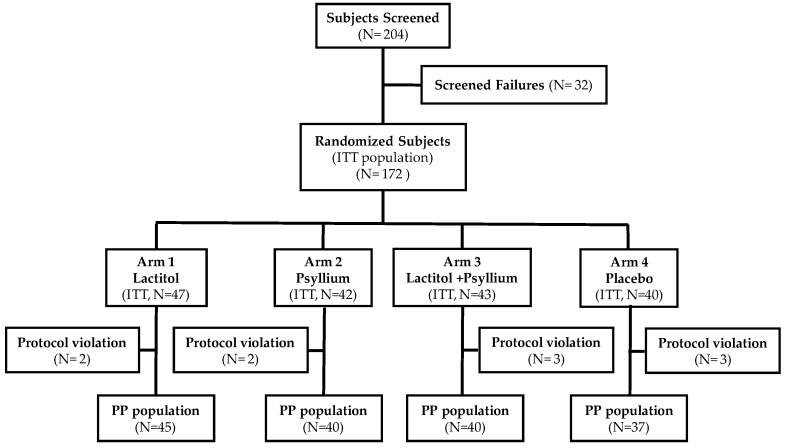
Disposition of volunteers. After the 2-week run-in period, 84% of volunteers were randomized into 4 treatment arms. Most randomized volunteers (94%) completed the trial. Withdrawals were evenly distributed among treatment groups. ITT—intention-to-treat; PP— per protocol.

**Table 1 nutrients-11-01130-t001:** Summary of Volunteer Demographics at Screening Visit for ITT Population (*N* = 172).

Category/Statistics, *n* (%) ^1^	Treatment Group				Overall (*N* = 172)
	Arm 1 (Lactitol, *N* = 47)	Arm 2 (Psyllium, *N* = 42)	Arm 3 (Lactitol + Psyllium, *N* = 43)	Arm 4 (Placebo, *N* = 40)	
Age ^2^					
Mean ± SD	39.8 ± 10.1	41.4 ± 11.1	40.1 ± 9.0	40.1 ± 12.1	40.3 ± 10.5
*p*-Value ^3^	0.91	0.6	1.0		
Gender					
Male	28 (59.6)	17 (40.5)	22 (51.2)	13 (32.5)	80 (46.5)
Female	19 (40.4)	25 (59.5)	21 (48.8)	27 (67.5)	92 (53.5)
*p*-Value ^3^	0.02 *	0.5	0.1		
Height (cm)					
Mean ± SD	161.5 ± 8.0	158.6 ± 7.3	162.2 ± 7.4	157.9 ± 8.9	160.1 ± 8.0
*p*-Value ^3^	0.046 *	0.7	0.02 *		
Weight (kg)					
Mean ± SD	59.7 ± 9.0	60.3 ± 7.1	61.0 ± 9.0	59.6 ± 7.1	60.2 ± 8.1
*p*-Value ^3^	1.0	0.6	0.4		
BMI					
Mean ± SD	22.8 ± 2.6	24.0 ± 2.3	23.1 ± 2.6	24.0 ± 2.6	23.5 ± 2.5
*p*-Value ^3^	0.046 *	1.0	0.2		

Note: *N*: number of subjects. ^1^ Respective column header group counts were used as denominator in the calculation of percentage. ^2^ Age calculated as: Age = ((Visit 1(screening) − Date of Birth + 1)/365.25). ^3^
*p*-value was calculated by *t*-test for continuous variables and chi-square test for categorical variable between active (lactitol, psyllium, and combination) vs placebo group. * *p* < 0.05 vs. placebo. Abbreviations: intention-to-treat (ITT), standard deviation (SD), body mass index (BMI).

**Table 2 nutrients-11-01130-t002:** Summary of Study Product Dispensing (IP Compliance) in ITT Population (*N* = 172).

IP Dispense	Treatment Group			
	Arm 1 (Lactitol, *N* = 47)	Arm 2 (Psyllium, *N* = 42)	Arm 3 (Lactitol + Psyllium *N* = 43)	Arm 4 (Placebo, *N* = 40)
Total sachets dispensed (*n*)	30	30	30	30
Total used sachets returned (*n*, mean ± SD)	27.2 ± 6.0	25.5 ± 8.2	26.3 ± 7.1	26.5 ± 6.3

Note: *N*: number of subjects. n: number of sachets. *p*-value was calculated by *t*-test between active (lactitol, psyllium, and combination) vs placebo group. Abbreviations: investigational product (IP), intention-to-treat (ITT), standard deviation (SD).

**Table 3 nutrients-11-01130-t003:** Summary of Actual value and Change from Baseline (V3) for main endpoints in ITT population.

Endpoints		Treatment Group											
		Arm 1 (Lactitol, *N* = 47)			Arm 2 (Psyllium, *N* = 42)			Arm 3 (Lactitol + Psyllium, *N* = 43)			Arm 4 (Placebo, *N* = 40)		
		Baseline	EOS	Δ ^2^	Baseline	EOS	Δ ^2^	Baseline	EOS	Δ ^2^	Baseline	EOS	Δ ^2^
BMF (BM/week) ^1^		2.4 ± 0.6	5.5 ± 2.1	3.0 ± 2.1 ^#^	2.3 ± 0.7	5.2 ± 1.8	2.9 ± 2.0 ^#^	2.3 ± 0.7	5.4 ± 2.0	3.1 ± 2.2 ^#^	2.3 ± 0.6	6.0 ± 3.1	3.7 ± 3.2 ^#^
BSFS ^1^		2.8 ± 1.0	3.6 ± 1.1	0.8 ± 1.2 ^#^	3.2 ± 1.1	3.8 ± 1.3	0.6 ± 1.0 ^#^	2.9 ± 1.1	3.7 ± 1.2	0.8 ± 1.5 ^#^	2.8 ± 1.1	3.8 ± 1.1	0.9 ± 1.0 ^#^
PAC-SYM ^1^	Abdominal	1.4 ± 0.8	0.5 ± 0.7	−1.0 ± 0.8 ^#^	1.4 ± 0.7	0.6 ± 0.8	−0.8 ± 0.7 ^#^	1.4 ± 0.7	0.6 ± 0.7	−0.8 ± 0.8 ^#^	1.5 ± 0.8	0.4 ± 0.5	−1.1 ± 0.9 ^#^
	Rectal	0.9 ± 0.6	0.3 ± 0.4	−0.6 ± 0.6 ^#^	0.9 ± 0.7	0.3 ± 0.6	−0.6 ± 0.7 ^#^	1.0 ± 0.6	0.3 ± 0.4	−0.8 ± 0.6 ^#^	0.8 ± 0.5	0.2 ± 0.4	−0.6 ± 0.5 ^#^
	Stool	2.2 ± 0.8	0.8 ± 0.7	−1.4 ± 0.8 ^#^	2.2 ± 0.8	1.0 ± 0.8	−1.3 ± 0.9 ^#^	2.2 ± 0.9	0.9 ± 0.8	−1.3 ± 1.0 ^#^	2.2 ± 0.8	0.8 ± 0.8	−1.4 ± 1.0 ^#^
	Total	1.6 ± 0.6	0.6 ± 0.6	−1.1 ± 0.6 ^#^	1.6 ± 0.6	0.7 ± 0.7	−1.0 ± 0.7 ^#^	1.6 ± 0.6	0.6 ± 0.6	−1.0 ± 0.7 ^#^	1.6 ± 0.6	0.5 ± 0.5	−1.1 ± 0.7 ^#^
PAC-QoL ^1^	Physical discomfort	2.2 ± 0.7	0.8 ± 0.9	−1.4 ± 0.9 ^#^	2.2 ± 0.6	0.8 ± 0.8	−1.4 ± 0.9 ^#^	2.2 ± 0.6	0.9 ± 0.9	−1.3 ± 0.8 ^#^	2.2 ± 0.7	0.7 ± 0.8	−1.4 ± 0.9 ^#^
	Psychosocial discomfort	1.9 ± 0.7	0.9 ± 0.8	−1.1 ± 0.9 ^#^	1.9 ± 0.5	0.8 ± 0.7	−1.1 ± 0.7 ^#^	1.9 ± 0.6	0.9 ± 0.8	−1.1 ± 0.8 ^#^	1.9 ± 0.6	0.8 ± 0.7	−1.1 ± 0.8 ^#^
	Worries and concerns	2.0 ± 0.7	0.9 ± 0.8	−1.1 ± 0.8 ^#^	1.9 ± 0.6	0.8 ± 0.9	−1.1 ± 0.8 ^#^	2.0 ± 0.6	1.0 ± 0.9	−1.0 ± 0.8 ^#^	2.0 ± 0.7	0.8 ± 0.8	−1.2 ± 0.8 ^#^
	Satisfaction	0.9 ± 0.4	2.0 ± 0.6	1.1 ± 0.7 ^#^	1.0 ± 0.4	2.0 ± 0.6	1.0 ± 0.7 ^#^	1.0 ± 0.3	2.0 ± 0.6	1.0 ± 0.7 ^#^	1.0 ± 0.4	2.0 ± 0.6	1.1 ± 0.7 ^#^
	Total	1.8 ± 0.6	1.1 ± 0.5	−0.7 ± 0.7 ^#^	1.8 ± 0.4	1.1 ± 0.6	−0.8 ± 0.6 ^#^	1.8 ± 0.5	1.1 ± 0.6	−0.7 ± 0.7 ^#^	1.8 ± 0.5	1.1 ± 0.7	−0.6 ± 0.6 ^#^
IPAQ (MET-minutes/ week) ^1^	Vigorous activity	2925.0 ± 4120.66	4237.1 ± 6922.1	2781.0 ± 7381.5	2226.7 ± 3354.4	2420.0 ± 4236.82	1000.0 ± 2107.97	6271.1 ± 7773.1	6680.0 ± 7245.29	−105.0 ± 3044.50	4692.0 ± 7370.3	5336.0 ± 7907.09	697.8 ± 2183.95
	Moderate activity	1843.3 ± 3429.5	2176.7 ± 4732.4	548.0 ± 1699.4	1970.0 ± 4037.2	2022.5 ± 4163.5	52.5 ± 703.05	2318.0 ± 2726.7	3000.0 ± 2664.5	682.0 ± 1218.3	1998.8 ± 2758.8	2530.0 ± 3800.0	530.5 ± 1695.7
	Walking activity	1856.0 ± 2640.3	1925.0 ± 2657.5	141.3 ± 837.2	1140.6 ± 1148.4	1289.8 ± 1157.6	129.5 ± 684.3	1611.1 ± 2188.4	1804.3 ± 2119.2	198.0 ± 831.3	1358.1 ± 1776.7	1861.4 ± 2253.0	526.7 ± 1548.3
	Total physical activity	2914.6 ± 5027.6	3541.2 ± 6859.9	774.0 ± 4105.9	1868.6 ± 3547.7	2156.6 ± 4515.9	290.0 ± 1323.1	3468.5 ± 6796.9	3748.0 ± 6642.4	286.3 ± 1780.4	3345.5 ± 7228.1	4124.1 ± 8254.9	816.2 ± 1935.5 *
ARC N (%) ^3^	Yes/No	Yes (7) No (40)	Yes (40) No (0)	Yes-> No 0(0) No-> Yes 33 (82.5)	Yes (3) No (39)	Yes (34) No (8)	Yes-> No 0(0) No-> Yes 31 (79.5)	Yes (6) No (37)	Yes (38) No (5)	Yes-> No 0(0) No-> Yes 32 (86.5)	Yes (7) No (33)	Yes (36) No (4)	Yes-> No 1 (14.3) No-> Yes 30 (90.9)
Overall product satisfaction N (%) ^4^	Very dissatisfied	-	0 (0.0)	-	-	0 (0.0)	-	-	2 (4.7)	-	-	0 (0.0%)	-
	Dissatisfied	-	4 (8.5)	-	-	5 (11.9)	-	-	4 (9.3)	-	-	5 (12.5)	-
	Neutral	-	8 (17.0)	-	-	7 (16.7)	-	-	7 (16.3)	-	-	5 (12.5)	-
	Satisfied	-	29 (61.7)	-	-	20 (47.6)	-	-	22 (51.2)	-	-	27 (67.5)	-
	Very satisfied	-	6 (12.8)	-	-	10 (23.8)	-	-	8 (18.6)	-	-	3 (7.5)	-

Note: ^1^ Values are mean ± SD. Paired *t*-test was applied for calculating the *p*-value in each treatment group comparing the EOS assessment with baseline results. ^2^ Δ: actual value change from baseline. ^3^ Count in Baseline column shows the number of patients at screening visit. Count of patients at EOS Visits and percentages are calculated using the Screening count as the denominator. ^4^ Percentages were calculated taking the respective column header count as the denominator. *p*-values were calculated using Pearson’s chi-square/Fisher’s exact test. In all comparisons (Δ) within each treatment group, *p*-values less than 0.05 are denoted by an asterisk (*), and *p*-values less than 0.01 labeled with a hash (^#^). Abbreviations: Adequate relief of constipation (ARC), intention-to-treat (ITT), end-of-study (EOS), bowel movement frequency (BMF), Bristol Stool Form Scale (BSFS), Patient Assessment of Constipation Symptoms (PAC-SYM), Patient Assessment of Constipation Quality of Life (PAC-QoL), International Physical Activity Questionnaire (IPAQ), standard deviation (SD).

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
