# Peer review of "Influence of Lactitol and Psyllium on Bowel Function in Constipated Indian Volunteers: A Randomized, Controlled Trial"

_nutrients, 2019, doi:10.3390/nu11051130_

Round 1
Reviewer 1 Report
Influence of lactitol and psyllium on bowel function in constipated Indian volunteers: A randomized, controlled trial
The authors assessed efficacy and safety of lactitol, psyllium and the combination of them compared to placebo (consisting of sucrose, instant waxy maize starch, citric acid, trisodium citrate, carotene powder and orange flavour), in a prospective, randomised study in adult volunteers with chronic constipation. They reported that all treatment groups showed comparable effects on all clinical endpoints however, the changes in BMF were shown in the Table 3 to be significantly better in the placebo group (p<0.01).
The study was well designed and of clinical relevance but the manuscript needs major revision as follows:
1. Study design: Can the authors explain from which cohort were the volunteers recruited? Was the Bristol scale taken into account at recruitment? Can the authors add information on the date of start and end of the recruitment?
2. Consort Flow Diagram: A figure with the Consort Flow Diagram is missing and should be added
3. Study design: On page 3, under the title of ‘Study design’ the methods of the study were described, The authors should separate ‘’Methods’’ from ‘’Study design’’.
2. Primary and secondary outcome measures: they were mentioned in the abstract but in the main body of the text they were not defined in a clear way. I would suggest to add for clarity two separate paragraphs describing the ‘’Primary’’ and ‘’Secondary’’ outcome measures respectively.
3. Registration at ClinicalTrials.gov: The study was an interventional study. Was it registered at ClinicalTrials.gov ? If yes, please add a statement with the Registration number and the date of the registration.
4. Results: The authors reported in the abstract that ‘’all clinical endpoints were similar in all treatment groups’’. In the Table 3 however, they reported better changes in BMF in the placebo group (p<0.01). This finding should be also, included in the abstract, with a brief explanation for that.
5. Discussion:
5a: The placebo in this study contained sucrose and instant waxy maize starch. Wang et al, had reported that banana resistant starch improved colonic transit in constipated mice (J Med Food. 2014 Aug;17(8):902-7). Would the authors like to commend on that? Could the composition of the placebo have an osmotic effect explaining its efficacy in treating constipation?
5b: The author included on page 7, line 323, the reference 44 saying that ‘’ However, consistent with our findings, Ewerth et al. 322 reported a nonsignificant decrease in BMF with psyllium compared with placebo [44].’’ The reference 44 (Acta Chir Scand Suppl. 1980;500:49-50) however, reported in (only) 9 patients with diverticuli of the colon and constipation who were treated with placebo and Vi-SiblinR in a double blind cross-over way, softer and increased in weight feces during the Vi-SiblinR treatment while the subjective symptoms were significantly reduced (p less than 0,05). Can the authors make a comment on that (explain or correct the reference in the Discussion to that citation)?
Tables: All of the abbreviations that are included in each table should be explained in full in the legends below the Tables.
Author Response
Reviewer 1: Major revision
General: Moderate English changes required.
Response: the text has been checked by a credited English language editor. We are attaching the certificate of the language proof-checking.
Specific: The authors assessed efficacy and safety of lactitol, psyllium and the combination of them compared to placebo (consisting of sucrose, instant waxy maize starch, citric acid, trisodium citrate, carotene powder and orange flavor), in a prospective, randomized study in adult volunteers with chronic constipation. They reported that all treatment groups showed comparable effects on all clinical endpoints, however, the changes in BMF were shown in the Table 3 to be significantly better in the placebo group (p<0.01). The study was well designed and of clinical relevance, but the manuscript needs major revision as follows:
1. Study design: Can the authors explain from which cohort were the volunteers recruited? Was the Bristol scale taken into account at recruitment? Can the authors add information on the date of start and end of the recruitment?
Response: this is from the patient population seeking care of bowel movement related complaints. Bristol scale was not taken into consideration at recruitment, while the BMF was considered at recruitment. The first sentence of the Result section 3 indicates the first participant in and out, as highlighted.
2. Consort Flow Diagram: A figure with the Consort Flow Diagram is missing and should be added
Response: in our opinion, the consort flow diagram was included in the whole submission file.
3. Study design: On page 3, under the title of ‘Study design’ the methods of the study were described, the authors should separate ‘’Methods’’ from ‘’Study design’’.
Response: the section was modified as suggested.
4. Primary and secondary outcome measures: they were mentioned in the abstract but in the main body of the text they were not defined in a clear way. I would suggest adding for clarity two separate paragraphs describing the ‘’Primary’’ and ‘’Secondary’’ outcome measures respectively.
Response: the section was modified as suggested.
5. Registration at ClinicalTrials.gov: The study was an interventional study. Was it registered at ClinicalTrials.gov ? If yes, please add a statement with the Registration number and the date of the registration.
Response: no. The study was registered in the Clinical Trials Registry for India (WHO approved Registry) with the registration identification number CTRI/2015/02/005580, as indicated in the text at Section 2.6. Ethical considerations.
6. Results: The authors reported in the abstract that ‘’all clinical endpoints were similar in all treatment groups’’. In the Table 3 however, they reported better changes in BMF in the placebo group (p<0.01). This finding should be also, included in the abstract, with a brief explanation for that.
Response: The reviewer is correct; a significant change was observed from baseline to the EOS within placebo group. However, similar changes were observed in the other treatment groups. No difference in change between groups noticed.
The legend for Table 3 was clarified to better represent what the superscript indicates.
7. The placebo in this study contained sucrose and instant waxy maize starch. Wang et al, had reported that banana resistant starch improved colonic transit in constipated mice (J Med Food. 2014 Aug;17(8):9027). Would the authors like to commend on that? Could the composition of the placebo have an osmotic effect explaining its efficacy in treating constipation?
Response: yes, it is correct. The resistant starch has influence on the colonic transit time (CTT) in constipated subjects. Therefore, we included in the placebo group, the waxy maize starch (WM) which should not have effects on the CTT, as it is easily digested. In addition, as stated in the discussion at page 8, the possibility of laxative effects from the ingredients in placebo was further ruled out in the post-hoc analysis.
8. The author included on page 7, line 323, the reference 44 saying that ‘’However, consistent with our findings, Ewerth et al. 322 reported a nonsignificant decrease in BMF with psyllium compared with placebo [44].’’ The reference 44 (Acta Chir Scand Suppl. 1980;500:4950) however, reported in (only) 9 patients with diverticuli of the colon and constipation who were treated with placebo and ViSiblinR in a double blind crossover way, softer and increased in weight feces during the ViSiblinR treatment while the subjective symptoms were significantly reduced (p less than 0,05). Can the authors make a comment on that (explain or correct the reference in the Discussion to that citation)?
Response: thanks for pointing out the inconsistency to us. We have corrected the section and highlighted.
Tables: All of the abbreviations that are included in each table should be explained in full in the legends below the Tables.
Response: all abbreviations are explained in full in the legends, as highlighted.
Reviewer 2 Report
I've read with attention the paper by Cheng et al., that is potentially of interest. The methodology applied is overall correct, the results are reliable and adequately discussed. I've only some minor comments:
- The abstract should include some quantiative data. At least the mean change in BMF.
- The lack of effects (even negative) of all tested products should be more deeply explained. In particular, I would expect that the increase in increase in fat intake and the decline in liquid intake seen in the placebo group should modifay the BMF... Do the authors think that the relatively large variability in the tested outcomes have made the study somewhat underpowered to detect differences among the secundary outcomes?
Author Response
Reviewer 2: Minor revision
General: English language and style are fine/minor spell check required
Specific: I've read with attention the paper by Cheng et al., that is potentially of interest. The methodology applied is overall correct, the results are reliable and adequately discussed. I've only some minor comments:
1. The abstract should include some quantitative data. At least the mean change in BMF.
Response: the quantitative data was included in the abstract and the section was shortened to comply with the length requirement of the journal, as highlighted.
2. The lack of effects (even negative) of all tested products should be more deeply explained. In particular, I would expect that the increase in fat intake and the decline in liquid intake seen in the placebo group should modify the BMF... Do the authors think that the relatively large variability in the tested outcomes have made the study somewhat underpowered to detect differences among the secondary outcomes?
Response: We agree with the reviewer but find difficult to explain the lack of difference between the treatments. Despite the variation in the dietary intakes, we had expected the increased intake in fiber (psyllium) and/or lactitol would have manifested itself as a change in BMF. Also, other studies that have investigated these products did not control dietary intake and observed changes in BMF. The study is indeed likely to be underpowered to investigate the influence of diet on the outcomes. For the secondary endpoints, the study is also likely to be insufficiently powered.
Round 2
Reviewer 1 Report
None